

# How well can persistent contrails be predicted? — An update

Sina Hofer[1], Klaus Gierens[1], and Susanne Rohs[2]

[1]Deutsches Zentrum für Luft- und Raumfahrt, Institut für Physik der Atmosphäre, Oberpfaffenhofen, Germany
[2]Forschungszentrum Jülich, IEK-8, Jülich, Germany

**Correspondence:** Sina Hofer (sina.hofer@dlr.de)

**Abstract.**

The total aviation effective radiative forcing is dominated by $non-CO_2$ effects. The largest contributors to the $non-CO_2$ effects are contrails and contrail cirrus. There is the possibility of reducing the climate effect of aviation by avoiding flying through ice supersaturated regions (ISSRs), where contrails can last for hours (so-called persistent contrails). Therefore, a

precise prediction of the specific location and time of these regions is needed. But a prediction of the frequency and degree of ice supersaturation (ISS) on cruise altitudes is currently very challenging and associated with great uncertainties because of the strong variability of the water vapour field, the low number of humidity measurements at air traffic altitude, and the oversimplified parameterisations of cloud physics in weather models.

Since ISS is more common in some dynamical regimes than in others, the aim of this study is to find variables/proxies that are

related to the formation of ISSRs and to use these for a regression method to predict persistent contrails. To find the best suited proxies for regressions, we use various methods of information theory. These include the log-likelihood ratios, known from the Bayes' theorem, a modified form of the Kullback-Leibler divergence and the mutual information. The variables (the relative humidity with respect to ice $RHi_{ERA5}$, the temperature $T$, the vertical velocity $\omega$, the divergence $DIV$, the relative vorticity $\zeta$, the potential vorticity $PV$, the normalised geopotential height $Z$ and the local lapse rate $\gamma$) come from ERA5 and $RHi_{M/I}$,

which we assume as the truth, comes from MOZAIC/IAGOS (commercial aircraft measurements).

It turns out, that $RHi_{ERA5}$ is the most important predictor of ice supersaturation, in spite of its weaknesses, and all other variables do not help much to achieve better results. Without $RHi_{ERA5}$, a regression to predict ISSRs is not successful. Certain modifications of $RHi_{ERA5}$ before the regression (as suggested in recent papers) do not lead to improvements of ISSR prediction. Applying a sensitivity study with artificially modified $RHi_{ERA5}$ distributions point to the origin of the problems

with the regression: the conditional distributions of $RHi_{ERA5}$ (conditioned on ISS and non-ISS, from $RHi_{M/I}$) overlap too heavily in the range 70-100%, such that for any case in that range it is not clear whether it belongs to an ISSR or not. Evidently, this renders the prediction of contrail persistence very difficult.

## 1 Introduction

In order to avoid persistent (warming) contrails, it is necessary that they can be reliably predicted. For this aim, three conditions

need to be fulfilled. First, the formation of contrails has to be predicted with a reasonable skill. Contrails form if (super-) saturation with respect to water occurs during the mixing process of the ambient air with the exhaust gases from the aircraft.



This criterion is called the Schmidt-Appleman criterion (SAC, see Schmidt, 1941; Appleman, 1953; Schumann, 1996). Second, contrails need ice supersaturation (ISS) to be persistent. So, this state must be represented and reliably predicted in weather models. Third, in order to determine in advance whether a contrail will be warming or cooling, some kind of radiative transfer
calculation or a corresponding regression formula (e.g., Corti and Peter, 2009; Schumann et al., 2012; Wolf et al., 2023a) is required.

While the first of these conditions, the ability to predict the SAC, is generally fulfilled with a satisfying quality, this is not the case for the prediction of ice supersaturation (Gierens et al., 2020). Predicting ice supersaturation at air traffic cruise levels presents major difficulties. Gierens et al. (2020) compared temperature and humidity data obtained from instrumented
passenger aircraft with reanalysis data interpolated in space and time to the measurement locations. They came to the result that currently the forecast of ice supersaturation at given times and locations (for flight routing purposes) is almost like tossing a coin. In contrast, the forecast of ISS is much easier for larger regions and periods of time (e.g. for planning measurement campaigns, see Voigt et al., 2017). In the present paper, we investigate the problem of the forecast of ISS in more detail and we will not cover the first and third condition.

There are several reasons why the prediction of persistent contrails is currently challenging. The main reason is the strong variability of the water vapour field in the atmosphere. This is because water substance is present in three aggregation states, it is involved in chemical and aerosol processes and thus it varies greatly in the atmosphere. This problem is intensified by the low number of humidity measurements at cruise levels for data assimilation. Data assimilation is necessary to keep the simulation of a complex system close to measured reality. Therefore, more data on relative humidity at flight levels are urgently needed.
Note that satellite data cannot fill this gap since their vertical resolution is insufficient (Gierens and Eleftheratos, 2020). A third reason for the challenging prediction of persistent contrails is that current parameterisations of ice cloud physics in weather models are generally kept simple enough in order to not spend too much computing time for a part of the atmosphere that so far was usually not the main focus of weather prediction. ISS hardly affects the weather on the ground. Thus, it was not represented in numerical weather prediction (NWP) models until about 25 years ago (Wilson and Ballard, 1999; Tompkins et al., 2007),
and its representation is still generally too crude for a reliable prediction of ice supersaturation and contrail persistence.

However, there is nowadays growing interest in reducing the climate impact of aviation, and a relatively straightforward possibility would be the avoidance of the formation of persistent contrails if only ice supersaturation could be predicted with the precision necessary for flight routing. Because of the challenges mentioned before, the relative humidity field is insufficient for this purpose and we need either corrections to the humidity field (Teoh et al., 2020, 2022) or other quantities that help
in the prediction of ISS. Gierens and Brinkop (2012) and Wilhelm et al. (2022) show that ISS is typically related to certain dynamical regimes, e.g. anticyclonic divergent uplift. This triggered the idea that using dynamical fields can indeed help in contrail forecast. We pursue this idea in the present study and show how far we can get with dynamical proxies together with modern regression methods. The results turn out to be considerably better than without such methods, but they are still not satisfying. A sensitivity study shows what impedes better results and where the source of the problem lies.

In the present paper, we concentrate on the prediction of ice supersaturation, respectively the prediction of persistent contrails. For this purpose, we use data obtained from instrumented passenger aircraft and reanalysis data, which are explained in



section 2. We test different methods for predicting persistent contrails, which is the content of section 3 as well as the results we obtained. Later on in section 4, we concentrate on modifying the relative humidity of ERA5 and on different sensitivity tests to artificially separate the conditional $RHi$-distributions of ERA5. Finally, we summarise our results and conclude in section 5.

## 2  Data

Various data sources are utilised in this study. These are briefly described in the following subsections 2.1 and 2.2.

### 2.1  Data from Commercial Aircraft

In this study we use pressure and relative humidity with respect to ice ($RHi_{\mathrm{M/I}}$) data collected from $16,588$ flights during 10 years (2000 - 2009) of MOZAIC (Measurement of Ozone and Water Vapour on Airbus In-service Aircraft, Marenco et al. (1998)) measurements. As MOZAIC was transferred to the European infrastructure IAGOS (In-service Aircraft for a Global Observing System, Petzold et al. (2015)) in 2011 we refer to the data as MOZAIC/IAGOS (www.iagos.org). MOZAIC/IAGOS operates autonomous in-situ instruments installed on long-range commercial aircraft. Due to this, it has the highest data density in the flight corridors of the mid-latitudes, making it an ideal data set for the investigation of contrail-forming regions. For this study we have chosen aerial boundaries of $30°$ to $70°$ latitude and $-125°$ to $145°$ longitude at cruise levels between 310 and 190 hPa, thus, a region with heavy air traffic. We use these data as a basic truth to determine whether the formation of a persistent contrail is possible or not at a specific position and time.

### 2.2  Reanalysis Data

In addition to the data from commercial aircraft, hourly ERA5 reanalysis data (Hersbach et al., 2018a, b, 2020) of the relative humidity with respect to ice $RHi_{\mathrm{ERA5}}$, the temperature $T$, the vertical velocity $\omega$, the divergence $DIV$, the relative vorticity $\zeta$, the potential vorticity $PV$ and the normalised geopotential height $Z$ for 200, 225, 250, and 300 hPa from ECMWF (European Centre for Medium-Range Weather Forecast) from the Copernicus Data Service (Copernicus Climate Change Service (C3S) (2017)) is retrieved and interpolated to the exact position and time of the aircraft. The chosen pressure range from 300 to 200 hPa covers approximately the flight levels of 300 to 390 hft. With the pressure and temperatures on two adjacent levels, also the local lapse rate $\gamma$ at the aircraft positions is calculated (Gierens et al., 2022). The selection of these particular variables comes from Wilhelm et al. (2022).

## 3  Methods and Results

In this work, we consider whether it is possible to use the dynamical proxies suggested by Wilhelm et al. (2022) to improve the forecast of ice supersaturation and contrail persistence. To quantify the success (or not) of several approaches we took, we use the equitable thread score, ETS, as in Gierens et al. (2020). The motivation for this choice of score value, the defining equations and interpretation of the ETS are given below in subsubsection 3.2.3.





The most simple way to map the values of the six dynamical proxies to probabilities for ice supersaturation or contrail persistence is to divide the phase space into 6-dimensional rectangles/blocks (6 because of the 6 suggested dynamical proxies by Wilhelm et al. (2022)), to count in each block the number of cases with persistent contrails and to divide it by the total number of data in that block. The blocks should not be too large so that the probabilities are specific for certain circumstances. Simultaneously, the blocks must not be too small, such that the number of events in each block allows one to determine the probability with some statistical reliability. Unfortunately, it turns out that even almost 400 thousand data points are insufficient for this simple and most straightforward method; many blocks are either empty or do not have enough data points unless we use quite large blocks and lose precision. Therefore, we cannot use this simple method and have to try others, like Bayesian learning or modern non-linear regression methods.

## 3.1 Bayesian Learning

### 3.1.1 Theory

We are interested in whether persistent contrails are possible or not, i.e. whether there is ice supersaturation or not. Unfortunately, the moisture field in the models is not accurate enough for that purpose. So, how can we solve this problem?

It is known that ISS is more frequent in some dynamic situations than in others (Gierens and Brinkop, 2012; Gierens et al., 2022; Wilhelm et al., 2022). One can try to exploit that: There are certain dynamical quantities $X$, whose conditional probability densities $f_X(x|ISS)$ and $f_X(x|\overline{ISS})$ differ more or less from each other (read: $f_X(x|ISS)$ as probability density for a quantity $X$ at the special value $x$ in cases where ISS prevails. $f_X(x|\overline{ISS})$ is the analogue for cases where non-ISS ($\overline{ISS}$) prevails.).

Let us assume there is a value $x$ of a variable $X$ and this particular value is compatible with both ISS and $\overline{ISS}$. Then the question arises: Which statements can be made about ice supersaturation using this quantity? Is it more likely or less likely when $X = x$?

Naively, one could compare $f_X(x|ISS)$ and $f_X(x|\overline{ISS})$ and choose the larger of the two values. However, this ignores that $\overline{ISS}$ cases are much more frequent than ISS cases when no further circumstances are considered, the so-called *a priori* probability. The latter is taken into account by Bayes' theorem in the following form:

$$P(ISS|X = x) = \frac{P(x|ISS) \cdot P(ISS)}{P(x|ISS) \cdot P(ISS) + P(x|\overline{ISS}) \cdot P(\overline{ISS})}, \tag{1}$$

where $P(ISS)$ is the a priori probability for ice supersaturation. $P(x|ISS) = f_X(x|ISS)\,\mathrm{d}x$ and $P(x|\overline{ISS}) = f_X(x|\overline{ISS})\,\mathrm{d}x$. In Equation 1, the $\mathrm{d}x$ are canceled out and on the right side of the equation, $P(x|ISS)$ and $P(x|\overline{ISS})$ can be replaced by the corresponding densities $f_X(x|ISS)$ and $f_X(x|\overline{ISS})$.

Another possibility to frame Bayes' theorem for the present problem is to use the odds ratio:

$$\frac{P(ISS|x)}{P(\overline{ISS}|x)} = \frac{f_X(x|ISS)}{f_X(x|\overline{ISS})} \cdot \frac{P(ISS)}{P(\overline{ISS})}. \tag{2}$$





The first factor on the right side of Equation 2 is the likelihood ratio, which represents the gain ($> 1$) or loss ($< 1$) in confidence for ISS that we get by learning the current value of $X$. A likelihood ratio exceeding 1 does not mean that ISS is more likely than $\overline{\text{ISS}}$, but the probability of ISS increases. ISS is only more likely than $\overline{\text{ISS}}$ if the factor on the left side, the a posteriori odds ratio, is $> 1$. The second factor on the right side is the prior odds ratio. In our case, $P(ISS)$ is given by the
125 ratio of the number of data with ISS and the total number of data, which is $0.115$. So, only $11.5\%$ of our data belong to the ISS class. The prior odds ratio is therefore $0.115/0.885 \approx 0.13$. This means that the likelihood ratio has to be larger than $3.85$ to make ISS more probable than $\overline{\text{ISS}}$. Instead of the odds ratio, one often uses its logarithm, which leads to log-likelihood ratios (also known as logits) with values symmetric around zero (instead of values asymmetric around one). That is, positive logits increase the probability for ISS and negative logits decrease the probability for ISS. As mentioned before, the a posteriori odds
ratio has to be $> 1$ to make ISS more probable than $\overline{\text{ISS}}$. This also means that the logarithm of the a posteriori ratio has to be positive to make ISS more likely than $\overline{\text{ISS}}$. As the logarithm of the prior odds ratio is $\ln(0.115/(1-0.115)) \approx -2$, the logit must exceed 2 to make ISS more probable than $\overline{\text{ISS}}$.

As long as there is only one special value $x$ of a dynamical variable $X$, we are finished and this is already the result. Now, assume that there are two quantities $X$ and $Y$ and one wants to know, which of these quantities carries more information about
135 the probability of ice supersaturation. Obviously, it is the quantity whose logits deviate stronger from zero in both the positive and negative directions, or the quantity for which the absolute values of the logits are larger on average over the ranges of $x$ and $y$. Taking the averages has to be done with a weighting that accounts for the values of the variables that actually occur in a given situation, e.g.

$$E_{\text{AL}}(f_{X|ISS}||f_{X|\overline{ISS}}) = \int f_X(x|ISS) \cdot \left| \ln \left( \frac{f_X(x|ISS)}{f_X(x|\overline{ISS})} \right) \right| \, \mathrm{d}x. \tag{3}$$

As one does not know in advance whether a situation is ISS or not, it is the best to also use the corresponding expectation of the absolute logit $E_{\text{AL}}(f_{X|\overline{ISS}}||f_{X|ISS})$ (where ISS and $\overline{\text{ISS}}$ are interchanged) and to average the two results. Let the result for $X$ be $E_{\text{AL}}(X)$. It may be called the expectation of the absolute logit. The quantity that yields the largest $E_{\text{AL}}(X)$ has the largest learning effect for the question of ISS or not.

Note that $E_{\text{AL}}(.||.)$ has some resemblance to a quantity known as Kullback-Leibler divergence $D_{\text{KL}}(f_{X|ISS}||f_{X|\overline{ISS}})$ (the
145 same expression without the absolute sign), and the corresponding symmetric form (the mean value of the two asymmetrical divergences $D_{\text{KL}}(f_{X|ISS}||f_{X|\overline{ISS}})$ and $D_{\text{KL}}(f_{X|\overline{ISS}}||f_{X|ISS})$) is known as Jeffries divergence in information theory. For quantities that are not related to supersaturation, $f_X(x|ISS) = f_X(x|\overline{ISS}) = f_X(x)$, and the logit is zero. Thus, $E_{\text{AL}}(X) = 0$ as well, which signifies that one cannot learn anything about the presence of ISS using such a quantity.

### 3.1.2 Application

The conditional probability densities of the dynamical proxies of ERA5 for ISS and $\overline{\text{ISS}}$ cases (depending on whether the relative humidity with respect to ice of MOZAIC/IAGOS $RHi_{\text{M/I}}$ is $>= 1.0$ or $< 1.0$) are calculated using the Epanechnikov smoothing kernel with 300 equally spaced points between the minimum and maximum of the respective proxy. The





log-likelihood ratios for some dynamical quantities are shown in Figure 1. ISS gains in probability relative to the low prior probability if the log-likelihood ratio is positive and vice versa (solid line in the diagrams). However, it needs to exceed 2 to make ISS more probable than $\overline{ISS}$ (dashed lines). Obviously, this threshold is only exceeded in quite small ranges of the proxies or not at all. Only where $RHi$ from ERA5 exceeds 100%, the logit exceeds 2; this says that ISS and persistent contrails are more probable than not (the wiggles in the curve at even higher $RHi$ are considered noise). The low values of the logits of the other variables indicate that the dynamical proxies do not help much in predicting ice supersaturation via the Bayesian law. Obviously, the strong separation of their conditional distributions is only a necessary but not a sufficient condition for good proxies.

For the calculation of the expectation of the absolute logit $E_{\mathrm{AL}}(X)$ (cf. Equation 3), the absolute values of the different logit functions are needed. These are shown in orange in Figure 2 for different proxies. The products of these functions with either of the conditional densities are shown as well in light blue and dark purple. The integrals of these functions are given in Table 1 for the relative humidity with respect to ice $RHi_{\mathrm{ERA5}}$, the temperature $T$, the vertical velocity $\omega$, the divergence $DIV$, the relative vorticity $\zeta$, the potential vorticity $PV$, the lapse rate $\gamma$ and the normalised geopotential height $Z$. The averages of the first two rows of each column are the desired $E_{\mathrm{AL}}(X)$. High values for $RHi_{\mathrm{ERA5}}$, $\zeta$, and $\gamma$ are noticeable for the ISS case, and high values for $RHi_{\mathrm{ERA5}}$, $PV$, and $\gamma$ for $\overline{ISS}$. Therefore, according to our analysis, $RHi_{\mathrm{ERA5}}$, $\zeta$, $\gamma$, and $PV$ seem to be the best-suited proxies for our purpose, but, as stated, they should be tried rather for regression and not for Bayesian learning. The high value of $E_{\mathrm{AL}}(PV)$ is probably due to the fact that a high $PV$ indicates the stratosphere where ISS hardly occurs. So, for tropospheric situations (low $PV$ cases) this finding is not very helpful, and accordingly, the high $E_{\mathrm{AL}}(PV)$ must not be over-interpreted.

To apply the Bayesian law for several proxies simultaneously, e.g. as for $P(ISS|RHi_{\mathrm{ERA5}},\zeta,\gamma)$, would need a much larger amount of data to compute the likelihood ratios with some robustness over the whole domain. Instead, we try now to apply non-linear regression.

**Table 1.** Expectation values for absolute logit of the different proxies. $E_{\mathrm{AL}}(.)$ is the mean of $E_{\mathrm{AL}}(f_{X|ISS}||f_{X|\overline{ISS}})$ and $E_{\mathrm{AL}}(f_{X|\overline{ISS}}||f_{X|ISS})$, so: $E_{\mathrm{AL}}(.) = \frac{1}{2}\cdot\Big(E_{\mathrm{AL}}(f_{X|ISS}||f_{X|\overline{ISS}}) + E_{\mathrm{AL}}(f_{X|\overline{ISS}}||f_{X|ISS})\Big)$.

| quantity | $RHi_{\mathrm{ERA5}}$ | $T$ | $\omega$ | $DIV$ | $\zeta$ | $PV$ | $\gamma$ | $Z$ |
|---|---|---|---|---|---|---|---|---|
| $E_{\mathrm{AL}}(f_{X|ISS}||f_{X|\overline{ISS}})$ | 1.75 | 0.34 | 0.56 | 0.42 | 0.96 | 0.90 | 0.96 | 0.52 |
| $E_{\mathrm{AL}}(f_{X|\overline{ISS}}||f_{X|ISS})$ | 3.41 | 0.38 | 0.43 | 0.33 | 1.20 | 2.07 | 1.24 | 0.88 |
| $E_{\mathrm{AL}}(.)$ | 2.58 | 0.36 | 0.50 | 0.38 | 1.08 | 1.49 | 1.10 | 0.70 |





**Figure 1.** Log-likelihood ratios for dynamical quantities. Positive values raise the probability for ISS, negative values lower it. The probability for ISS exceeds the probability for $\overline{\text{ISS}}$ only in the small ranges where the log-likelihood reaches values above 2 (marked by the dashed lines).



**Figure 2.** The absolute logarithm of the quotient of the densities for ISS and $\overline{\text{ISS}}$ cases (orange), the absolute logarithm of the quotient of the densities for ISS and $\overline{\text{ISS}}$ cases times the density for ISS cases (light blue), and the absolute logarithm of the quotient of the densities for $\overline{\text{ISS}}$ and ISS cases times the density for $\overline{\text{ISS}}$ cases (dark purple) (cf. Equation 3).



## 3.2 Non-linear Regression

The dynamical candidate proxies are not independent quantities, and one has to take care that a regression is not formulated with redundant information. But, of course, a variable that has some relation with (i.e. information on) the relative humidity is welcome. Above we have seen that $RHi_{\mathrm{ERA5}}$, $\zeta$, and $\gamma$ are promising in this respect. Also, $PV$ has a quite large absolute logit, but that comes mainly from the $\overline{\mathrm{ISS}}$ cases, where it is an expression of the fact that dry stratospheric air (with $PV > 3.5$) is rarely found in a supersaturated state (Neis, 2017; Petzold et al., 2020). Here we will apply another measure. Usually, one uses the linear correlation between the input data, but this does not work if the quantities are related in a non-linear fashion. Therefore, it is necessary to use a more general measure of correlation, namely the mutual information from information theory.

### 3.2.1 Mutual Information

The mutual information is a measure of information that one variable, $X$, can provide about another, $Y$. Its formulation uses the joint distribution of $X$ and $Y$ and both marginal distributions:

$$I(X;Y) = \iint f_{X \cap Y}(x,y) \cdot \log\left(\frac{f_{X \cap Y}(x,y)}{f_X(x) \cdot f_Y(y)}\right) \mathrm{d}x\,\mathrm{d}y, \tag{4}$$

where $f_{X \cap Y}(x,y)$ is the joint probability density for $X$ and $Y$. In case that $X$ and $Y$ are independent, the joint density equals the product of the marginal densities and the logarithm is zero. Then the mutual information between $X$ and $Y$ is zero as well. In all other cases, it is positive and it is the expected value of $\log(\frac{f_{X \cap Y}}{f_X f_Y})$ with the joint distribution as the weight function.

Since we assume the humidity of MOZAIC/IAGOS ($RHi_{\mathrm{M/I}}$) is the truth, we calculate the mutual information of $RHi_{\mathrm{M/I}}$ with other quantities and compare them with each other. The results of the computed mutual information $I(RHi_{\mathrm{M/I}};Y)$ for different variables $Y$ are shown in the first row in Table 2. The highest values of the mutual information are reached by $RHi_{\mathrm{ERA5}}$ (1.26 bits), PV (0.57 bits), $\gamma$ (0.38 bits) and $\zeta$ (0.37 bits). This means, according to the mutual information, $RHi_{\mathrm{ERA5}}$, $PV$, $\gamma$ and $\zeta$ seem well-suited as proxies for regressions.

To be a good proxy for a regression, it must not only be well correlated with $RHi_{\mathrm{M/I}}$ (i.e. have a high value of mutual information $I(RHi_{\mathrm{M/I}};Y)$), but at the same time, it should not be correlated with other variables (i.e. a low value of mutual information with the other proxies $I(X;Y)$) to avoid redundant information. The values of the individual mutual information $I(X;Y)$ can be found on the right side in Table 2. The matrix is symmetrical, so for a better overview only one side is filled .

As mentioned before, from all quantities, $RHi_{\mathrm{ERA5}}$, $PV$, $\gamma$ and $\zeta$ have the highest mutual information with $RHi_{\mathrm{M/I}}$. However, $PV$ and $\gamma$ are themselves quite strongly correlated (1.07 bits). So, because $I(RHi_{\mathrm{M/I}};PV) > I(RHi_{\mathrm{M/I}};\zeta)$, but $I(PV;\zeta)$ is also very high, $\zeta$ should be omitted when using $PV$ as a proxy.





**Table 2.** Mutual information matrix $I(X;Y)$ to identify the correlations of the variables with each other. The first row of the matrix, $I(RHi_{M/I};Y)$, shows the correlation of $RHi_{M/I}$ and the other proxies and the columns the correlations between the variables among themselves.

| $I(X;Y)$ in bits | $RHi_{M/I}$ | $RHi_{ERA5}$ | $T$ | $\omega$ | $DIV$ | $\zeta$ | $PV$ | $\gamma$ | $Z$ |
|---|---|---|---|---|---|---|---|---|---|
| $RHi_{M/I}$ | | 1.26 | 0.29 | 0.06 | 0.04 | 0.37 | 0.57 | 0.38 | 0.23 |
| $RHi_{ERA5}$ | | | 0.35 | 0.08 | 0.06 | 0.43 | 0.61 | 0.43 | 0.25 |
| $T$ | | | | 0.02 | 0.02 | 0.12 | 0.16 | 0.05 | 0.36 |
| $\omega$ | | | | | 0.21 | 0.04 | 0.09 | 0.10 | 0.06 |
| $DIV$ | | | | | | 0.03 | 0.04 | 0.04 | 0.03 |
| $\zeta$ | | | | | | | 0.73 | 0.43 | 0.19 |
| $PV$ | | | | | | | | 1.07 | 0.54 |
| $\gamma$ | | | | | | | | | 0.54 |
| $Z$ | | | | | | | | | |

### 3.2.2 Generalised Additive Model

A generalised additive model (GAM) is a regression method for predicting a response $Y$ based on non-linear functions of several predictors $X = (X_1, X_2, ..., X_p)$. In meteorology, for instance, it has been successfully used for the prediction of thun-
derstorms (Rädler et al., 2018). The general formula for GAMs is:

$$\log\left(\frac{P(ISS|X)}{1 - P(ISS|X)}\right) = \beta_0 + s_1(X_1) + s_2(X_2) + ... + s_p(X_p). \tag{5}$$

$\frac{P(ISS|X)}{1-P(ISS|X)}$ is the posterior odds ratio $P(ISS|X)/P(\overline{ISS}|X)$. The GAM thus constructs a relation between the (posterior) odds ratio and a linear combination of functions of the predictors $X$. For the functions we use smoothing splines $s(X)$. Here, we test six different GAMs with combinations of various dynamical proxies (input parameters) to predict whether persistent
contrails are possible:

 

- GAM$_0$ with $RHi_{ERA5}$,

- GAM$_1$ with $T, RHi_{ERA5}$,

- GAM$_2$ with $PV, T, RHi_{ERA5}$,

- GAM$_3$ with $PV, T, \zeta, RHi_{ERA5}$,

- GAM$_4$ with $\gamma, T, Z, PV, \zeta$, and

- GAM$_5$ with $\gamma, T, Z, PV, \zeta, RHi_{ERA5}$ .



The procedure is as follows: For the tests, $395,576$ independent data points are used. First, we divide the data set (MOZA-IC/IAGOS and ERA5 data) into a training and test data set (training data set $\sim 80\%$ and test data set $\sim 20\%$ of the whole

data set). The presence of persistent contrails is known from the MOZAIC/IAGOS data, as described in subsection 2.1. Next, we train the model, which means, we find the best coefficients for the relationships between the proxies and prediction using the training data set and a software environment for statistical computing called $R$ (R. Ihaka and R. Gentleman, 1993). Then we use the gained best coefficients and functions to predict the presence of persistent contrails in the test data set. At the end, we validate the forecast by comparing the predictions with the truth and calculating the so-called equitable threat score (see

subsubsection 3.2.3).

### 3.2.3 Equitable Threat Score

In this study, the equitable threat score (ETS) is used to validate and compare the prediction accuracy of the different GAMs (with varied input parameters) and of the raw data. For the calculation of the ETS (Gierens et al., 2020), the events are summed up according to the contingency table (see Table 3), where a distinction is made between "potential persistent contrails

predicted" yes/no and "persistent contrails observed" yes/no.

**Table 3.** Contingency table for predicting and observing persistent contrails.

|  |  | potential persistent contrails predicted | |
| --- | --- | --- | --- |
|  |  | yes | no |
| persistent contrails observed | yes | correct ($a$) | false negative ($b$) |
|  | no | false positive ($c$) | correct ($d$) |

The sum of the events is labeled as $a$ (contrails are predicted and observed), $b$ (no contrails are predicted but observed), $c$ (contrails are predicted but not observed), and $d$ (contrails are neither predicted nor observed). For the calculation of the ETS, the numbers of the events $a$, $b$, $c$, and $d$ and the following equation are used:

$$ETS = \frac{a - r}{a + b + c - r} \tag{6}$$

with

$$r = \frac{(a + b) \cdot (a + c)}{a + b + c + d}. \tag{7}$$

If the prediction agrees perfectly with the observation, ETS $= 1$. For a completely inverse relation, ETS is negative, and for

a random relation, ETS $= 0$. The advantage of using ETS instead of another skill score is that the influence of a predominant no/no case (large value of $d$) is minimised, which we have here, since $\overline{\text{ISS}}$ is much more probable than ISS.





In order to fill the contingency table, it is necessary to decide on a conditional probability threshold $P(ISS|X)$ up to which $\overline{ISS}$ and from which on ISS is predicted. To determine the threshold, one generally uses the value that gives the best ETS. In the present case, this threshold probability is $0.34$. That is, with a given vector $X$ of proxies, we predict contrail persistence or
ice supersaturation if $P(ISS|X) > 0.34$ (and vice versa).

### 3.2.4 Regression Results

Table 4 shows the results of the application of six different general additive models ($\text{GAM}_0$ to $\text{GAM}_5$). Note that $\beta_0$ is different for every GAM even if it is abbreviated in the same way in all GAMs. In the first (white) row, the ETS value of $0.198$ is given, which is obtained when only $RHi_{\text{ERA5}}$ and $RHi_{\text{M/I}}$ are compared (without applying a GAM to it). In that case, we only
check how well the prediction of ISS in ERA5 matches the observation of ISS of MOZAIC/IAGOS. So, it is only examined how often $RHi_{\text{ERA5}} >= 1.0$ matches $RHi_{\text{M/I}} >= 1.0$. The SAC is not taken into account.

In $\text{GAM}_0$, we only use $RHi_{\text{ERA5}}$ and get an ETS value of $0.337$. In $\text{GAM}_1$, we also take $T$ into account, since the temperature and the humidity are the two important variables to compute whether contrails form (using the SAC). With these proxies we get an ETS of $0.372$, which is a little bit higher than in $\text{GAM}_0$, but not significantly.

As we saw in subsection 3.2.1, when calculating the mutual information, $RHi_{\text{ERA5}}$, $PV$, $\zeta$ and $\gamma$ show particularly high values with $RHi_{\text{M/I}}$ ($I(RHi_{\text{M/I}}; RHi_{\text{ERA5}}) = 1.26$ bits, $I(RHi_{\text{M/I}}; PV) = 0.57$ bits, $I(RHi_{\text{M/I}}; \gamma) = 0.38$ bits and $I(RHi_{\text{M/I}}; \zeta) = 0.37$ bits) and are therefore very suitable as proxies. But when looking at the mutual information among these proxies, then it is noticeable that in particular $PV$ and $\gamma$ correlate strongly ($1.07$ bits). That is why $\gamma$ can be omitted when using $PV$. $I(PV; \zeta)$ is also very high ($0.73$ bits), which is why we only use the $PV$ in $\text{GAM}_2$ (in addition to $T$ and $RHi_{\text{ERA5}}$). The
resulting ETS value is also $0.372$.

For $\text{GAM}_3$ we use $PV$, $T$, $\zeta$ and $RHi_{\text{ERA5}}$ because we want to take even more proxies into the GAM as inputs according to the mutual information. So, we do the same GAM as before but we also use $\zeta$, because $I(RHi_{\text{M/I}}; \zeta)$ is also very high and $I(PV; \zeta) < I(PV; \gamma)$. The ETS in this case is $0.373$. We see, that $\text{GAM}_2$ and $\text{GAM}_3$ hardly differ from $\text{GAM}_1$ in terms of their ETS values. The reason for this is, even if $I(RHi_{\text{M/I}}; PV)$ is very high, the $PV$ is already very strongly correlated with
$RHi_{\text{ERA5}}$ ($I(RHi_{\text{ERA5}}; PV) = 0.61$ bits).This means that not much more additional information is provided by $PV$ (and the other variables even provide less).

Next, we use all proxies that show separate distributions in their probability density functions (PDFs, not shown) $P(X|ISS)$ vs. $P(X|\overline{ISS})$, but, as an experiment, we omit $RHi_{\text{ERA5}}$ in $\text{GAM}_4$. So, we use as inputs: $\gamma$, $T$, $Z$, $PV$ and $\zeta$. The ETS only reaches a value of $0.197$. This shows us two important things:

i) Even if we supposedly put more information into the GAM by using more proxies, the ETS does not increase.

ii) The relative humidity can not be ignored as an input variable. This indicates that even if the relative humidity is an imprecise variable, it must not be excluded, otherwise, the ETS value will drastically decrease.

These two new insights may be explained by the log-likelihood ratios (Figure 1): Only $RHi_{\text{ERA5}}$ shows values above $2$ for a large range, so ISS is more probable in this range. This is probably the reason why $RHi_{\text{ERA5}}$ has to be used as an input for the
GAMs. All other quantities either show no values above $2$ at all or only for a very small range.





By using the same proxies as before and adding $RHi_{\mathrm{ERA5}}$ (GAM$_5$), the corresponding ETS reaches a value of 0.378.

It seems that the use of dynamical proxies in the GAMs does not outperform by much a simple GAM that uses only relative humidity and temperature. At least, the ETS values obtained via the GAMs (that is, for prediction of potential persistent contrails) distinctively exceed those obtained from a simple check of the ISS prediction, as from the study of Gierens et al. (2020, ETS $= 0.08$ for a relatively small set of data from 2014) and from the present much larger data set (ETS $= 0.198$). This means that even if a value of 0.378 seems small at first glance, it is still larger than if the prediction of ice supersaturation is purely based on the relative humidity with respect to ice from the ERA5 data.

Note that despite $T$ is not particularly prominent, neither in the $E_{\mathrm{AL}}$ nor in its mutual information with $RHi_{\mathrm{M/I}}$, we use it in GAM$_1$ to GAM$_5$ because it is such an important quantity in the SAC and we use the proxies to provide further information (in addition to $T$ and $RHi_{\mathrm{ERA5}}$). So, when running our best GAM (GAM$_5$), but this time without $T$, the ETS reaches a value of 0.357. $T$ does not increase the ETS significantly, but we leave it in for the reasons mentioned above.

Since, as we have seen, the relative humidity should definitely be used as an input for a GAM, although it is not very precise, the questions arise: Is it possible to improve the regression results using corrections to the relative humidity field from the weather forecast models and what is the reason why even the most advanced regression methods are not able to yield better ETS values? These questions are dealt with in the next section.

**Table 4.** Results of comparing $RHi_{\mathrm{ERA5}}$ and $RHi_{\mathrm{M/I}}$ with each other and of different GAMs.

| Comparison of raw data (assessment of the ISS-prediction; without using a GAM) | | ETS |
|---|---|---|
| $RHi_{\mathrm{ERA5}}$ and $RHi_{\mathrm{M/I}}$ | | 0.198 |
| **Prediction of potential persistent contrails using proxies and GAMs:** $\log\left(\frac{p(X)}{1-p(X)}\right) =$ | | **ETS** |
| GAM$_0$ | $\beta_0 + s(RHi_{\mathrm{ERA5}})$ | 0.337 |
| GAM$_1$ | $\beta_0 + s(T) + s(RHi_{\mathrm{ERA5}})$ | 0.372 |
| GAM$_2$ | $\beta_0 + s(PV) + s(T) + s(RHi_{\mathrm{ERA5}})$ | 0.372 |
| GAM$_3$ | $\beta_0 + s(PV) + s(T) + s(\zeta) + s(RHi_{\mathrm{ERA5}})$ | 0.373 |
| GAM$_4$ | $\beta_0 + s(\gamma) + s(T) + s(Z) + s(PV) + s(\zeta)$ | 0.197 |
| GAM$_5$ | $\beta_0 + s(\gamma) + s(T) + s(Z) + s(PV) + s(\zeta) + s(RHi_{\mathrm{ERA5}})$ | 0.378 |

## 4  Sensitivity Tests

If weather forecasts were perfect, contrail persistence could easily be predicted using temperature and relative humidity alone and it would not be necessary to use any proxies. Unfortunately, it seems that in particular the predicted humidity field (at least from ERA5, but certainly from other weather models as well) is not good enough to allow such a forecast for single flights, that is, waypoint to waypoint (Gierens et al., 2020). There are plausible reasons for this, in particular a lack of in-situ observations of humidity at cruise levels and outdated cirrus parameterisations in numerical weather prediction models (Sperber and Gierens, 2023). In the following we perform regression tests with artificially changed distributions of relative





humidity. We first assume that the two conditional $RHi_{\text{ERA5}}$ distributions $P(RHi_{\text{ERA5}}|ISS)$ and $P(RHi_{\text{ERA5}}|\overline{ISS})$ were more separated (less overlap) than they are (ideally the overlap should be very small, including only sublimating contrails in

the ISS-conditioned PDF and supersaturated, but too warm cases in the $\overline{\text{ISS}}$-conditioned PDF). Second, we test two different methods of humidity corrections to see whether they help to reach higher ETS values in the regression models.

## 4.1   Separating the probability density functions conditioned on persistence

We guess that the root of the problem of predicting ice supersaturation and contrail persistence is the too strong overlap of the two conditional humidity PDFs, namely $f_{RHi_{\text{ERA5}}}(r|ISS)$ and $f_{RHi_{\text{ERA5}}}(r|\overline{ISS})$, where $r$ is a special value of $RHi_{\text{ERA5}}$.

This substantial overlap can be seen in panel f) of Figure 3. Now we artificially separate these two distributions using a perfectly separated pair of distributions, a log-normal distribution $f_{\text{PC}}$ cut-off at $0.8$ and $1.5$ for cases that allow persistent contrails (PC), and a second one, $f_{\text{noPC}}$ ranging from $0.0$ and $0.8$ for cases that do not (no PC). Then we mix the original conditional probability distributions of all data, both in the training and test data set, more and more into the artificial distributions, namely using a weighting factor $0 \leq a \leq 1$ as follows:

$$
\begin{aligned}
f(r|\overline{ISS}, a) &= a f_{\text{noPC}} + (1-a) f_{RHi_{\text{ERA5}}}(r|\overline{ISS}), \\
f(r|ISS, a) &= a f_{\text{PC}} + (1-a) f_{RHi_{\text{ERA5}}}(r|ISS).
\end{aligned}
\tag{8}
$$

Some examples of these artificial distributions are shown in Figure 3. From these distributions, we draw random humidity values and replace the original ones with them keeping their ISS and $\overline{\text{ISS}}$ label (i.e. either persistent contrail, $RHi_{\text{a}}$ drawn from $f(r|ISS, a)$, or non-persistent or no contrail drawn from $f(r|\overline{ISS}, a)$). This data set is then again divided into a training

(80%) and test data (20%) set and GAMs (with $T$ and $RHi_{\text{a}}$, like in $\text{GAM}_1$) and the corresponding ETS are computed for every value of $a$.

   The results are shown in Table 5. It is noticeable that even with a small shift in the relative humidity data, the ETS value increases drastically. This can be observed especially for small values of $a$, which means that if the original $RHi_{\text{ERA5}}$ data were just slightly more separated, the results would be drastically improved. This is good news since it shows that the model

prediction of $RHi_{\text{ERA5}}$ does not need to be perfect. Very good ETS values already appear for conditional distributions that are a little less separated than they actually are. Note that the further increase of ETS for $a > 0.5$ is quite weak since ETS $(a = 0.5)$ already exceeds $0.9$.





**Figure 3.** Conditional probability density functions $f(r|ISS, a)$ (blue) and $f(r|\overline{ISS}, a)$ (red) for different values of $a$. Note that the original distributions $f_{RHi_{\mathrm{ERA5}}}(r|\overline{ISS})$ and $f_{RHi_{\mathrm{ERA5}}}(r|ISS)$ are retained with $a = 0.0$ in panel f).

**Table 5.** Results of the sensitivity test

| $a$ | 1.0 | 0.9 | 0.8 | 0.7 | 0.6 | 0.5 | 0.4 | 0.3 | 0.2 | 0.1 | 0.0 |
|-----|-----|-----|-----|-----|-----|-----|-----|-----|-----|-----|-----|
| **ETS** | 1.0 | 0.996 | 0.989 | 0.976 | 0.956 | 0.921 | 0.863 | 0.765 | 0.639 | 0.484 | 0.372 |



## 4.2 Correction Formulas

As subsection 4.1 shows, there is a strong increase in the ETS for decreasing overlap. For this reason, two different methods are
being tested to further separate the two conditional distributions, $f_{RHi_{ERA5}}(r|ISS)$ and $f_{RHi_{ERA5}}(r|\overline{ISS})$, using corrections
to the modelled relative humidity values. These correction methods are quantile mapping based on the present data sets (e.g.
Gierens and Eleftheratos, 2017; Wolf et al., 2023b) and the *RHi*-modification used by Teoh et al. (2022).

### 4.2.1 Quantile Mapping

The quantile mapping procedure uses the two cumulative distributions of $RHi$, the one from the MOZAIC/IAGOS data and
the corresponding one from ERA5, see Figure 4. Evidently, the two distributions differ, in particular around saturation. This
is therefore the range of values, where corrections have the greatest effect. The procedure is quite simple: for each $RHi_{ERA5}$
the corresponding quantile value (the value on the $y$-axis) is looked up and the the corresponding $RHi_{M/I}$, that has the same
quantile value, is taken as the corrected $RHi_{QM}$. This is illustrated by the black arrows in Figure 4. We note that saturation
($RHi_{QM} = 1$) is already reached at the predicted (i.e. ERA5) relative humidity of $RHi_{ERA5} = 0.934$, using this method.

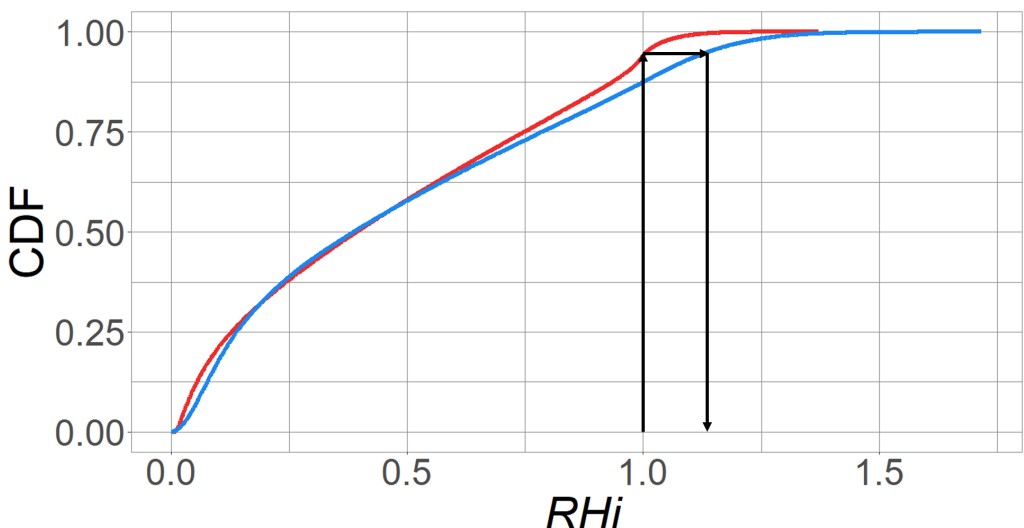

**Figure 4.** Illustration of quantile mapping for $RHi_{ERA5}$. The red curve illustrates the cumulative distribution function (CDF) of $RHi_{ERA5}$
and the blue one the CDF of $RHi_{M/I}$.

### 4.2.2 *RHi*-modification by Teoh et al. (2022)

In a study by Teoh et al. (2022), the ERA5 ice supersaturation over the North Atlantic was adjusted to the corresponding
MOZAIC/IAGOS supersaturation by introducing two factors with whom the $RHi_{ERA5}$ is scaled:





$$
\begin{aligned}
RHi_{\text{TEOH}} &= \frac{RHi_{\text{ERA5}}}{a} \qquad \text{if} \qquad \frac{RHi_{\text{ERA5}}}{a} \leq 1 \\
RHi_{\text{TEOH}} &= \min\left[\left(\frac{RHi_{\text{ERA5}}}{a}\right)^b, 1.65\right] \qquad \text{if} \qquad \frac{RHi_{\text{ERA5}}}{a} > 1,
\end{aligned}
\tag{9}
$$

with $a = 0.9779$ and $b = 1.635$. Here we try whether this modification can lead to improvements in our regression models.

### 4.2.3   Results using corrections

Table 6 shows the results of comparing the observed ice supersaturation $RHi_{\text{M/I}}$ with the modified relative humidities with respect to ice, $RHi_{\text{QM}}$ and $RHi_{\text{TEOH}}$, and the results of using these corrected humidities in the best GAM we have found before (see subsubsection 3.2.4). To recall the results to be compared, which we have already described, the ETS of the

comparison of $RHi_{\text{ERA5}}$ and $RHi_{\text{M/I}}$ and also the original $\text{GAM}_5$ have been added to the table.

We check how good the prediction of ice supersaturation is, using the corrected versions of $RHi_{\text{ERA5}}$. When comparing the data of $RHi_{\text{M/I}}$ with the modified relative humidity with respect to ice using the quantile mapping method, $RHi_{\text{QM}}$, the ETS reaches a value of $0.344$. If the relative humidity with respect to ice is modified according to the formula of Teoh $RHi_{\text{TEOH}}$ and compared to $RHi_{\text{M/I}}$, than the ETS is $0.284$.

**Table 6.** Results of comparing $RHi_{\text{M/I}}$ and the modified humidities $RHi_{\text{QM}}$ and $RHi_{\text{TEOH}}$ and of GAMs using $RHi_{\text{QM}}$ and $RHi_{\text{TEOH}}$.

| Comparison of raw data (assessment of the ISS-prediction; without using a GAM) | ETS |
|---|---|
| $RHi_{\text{ERA5}}$ and $RHi_{\text{M/I}}$ | 0.198 |
| $RHi_{\text{QM}}$ and $RHi_{\text{M/I}}$ | 0.344 |
| $RHi_{\text{TEOH}}$ and $RHi_{\text{M/I}}$ | 0.284 |
| **Prediction of potential persistent contrails** using proxies and GAMs: $\log\left(\frac{p(X)}{1-p(X)}\right) =$ | **ETS** |
| $\text{GAM}_5$     $\beta_0 + s(\gamma) + s(T) + s(Z) + s(PV) + s(\zeta) + s(RHi_{\text{ERA5}})$ | 0.378 |
| $\text{GAM}_{5,\text{QM}}$     $\beta_0 + s(\gamma) + s(T) + s(Z) + s(PV) + s(\zeta) + s(RHi_{\text{QM}})$ | 0.377 |
| $\text{GAM}_{5,\text{TEOH}}$     $\beta_0 + s(\gamma) + s(T) + s(Z) + s(PV) + s(\zeta) + s(RHi_{\text{TEOH}})$ | 0.376 |

Now, we use the same proxies as in $\text{GAM}_5$ but we replace $RHi_{\text{ERA5}}$ with $RHi_{\text{QM}}$ gained by quantile mapping, and for the other case with $RHi_{\text{TEOH}}$ using the formula by Teoh. The relative humidity of the whole data set is adapted. When $RHi_{\text{ERA5}}$ is modified by quantile mapping, we get an ETS value of $0.377$ and for a change in humidity according to Teoh, the ETS value is $0.376$.

Unfortunately, it turns out that neither a GAM produced with quantile-mapped ERA5 humidity values nor a GAM where the

Teoh et al. (2022) corrections have been applied, leads to larger ETS values than a GAM without the corrections (reminder: the ETS of the original $\text{GAM}_5$ with the original $RHi_{\text{ERA5}}$ is $0.378$). In comparison to the ETS of $0.344$ (quantile mapping)

and 0.284 (Teoh) mentioned above, when only the modified humidities are compared with $RHi_{\mathrm{M/I}}$, the GAMs with the best suited proxies and the modified humidities affect the ETS values hardly.

The probable reason for this negative result is seen in Figure 5, which shows the original conditional PDFs (red and blue) together with those obtained when the corrections are applied. Evidently, there is some shift in particular for the PDFs conditioned on ISS, but the overlap between the distribution pairs still remains considerable, too substantial for a better result.

Another reason for the insensitivity of the GAMs to these corrections may be that they absorb such modifications in the coefficients of the non-linear smooth functions. This may become clearer, if one thinks of a linear regression ($Y = \beta_0 + \beta_1 X + \varepsilon$) where a linear correction of the predictor $X$, that is $X' = a + bX$, would also be simply absorbed in the regression coefficients $\beta_0, \beta_1$. That is, they would simply take different values but the form of the regression and the ETS would not change.

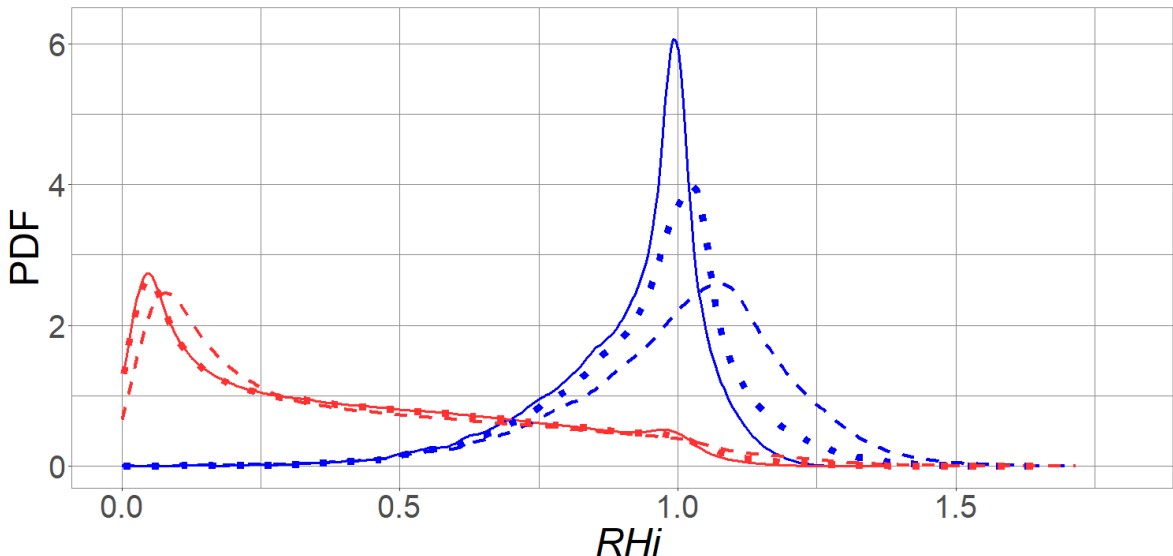

**Figure 5.** The conditioned PDFs $P(RHi|ISS)$ (blue) and $P(RHi|\overline{ISS})$ (red) of different *RHi* distributions are shown. The solid curves are the PDFs of the original $RHi_{\mathrm{ERA5}}$ of the whole data set for $\overline{ISS}$ and ISS. The other curves represent *RHi*-corrections: The dashed lines are the conditioned PDFs of $RHi_{\mathrm{QM}}$ modified by the quantile mapping method. The dotted curves represent the conditioned PDFs of $RHi_{\mathrm{TEOH}}$ modified by the Teoh formula.

## 5 Summary and Conclusions

There are various approaches to minimise the climate impact of aviation. One of these approaches is to prevent the formation of persistent contrails by avoiding flying through ice supersaturated regions where contrails can last for hours. For implementing such aircraft diversions, these regions have to be predicted accurately in terms of time and location, which is currently associated with difficulties and uncertainties. This is mainly due to the inaccurate forecast of the relative humidity. Since ice



supersaturation (ISS) is more common in some dynamical regimes than in others, we use different dynamical proxies (in addition to the relative humidity with respect to ice $RHi_{\mathrm{ERA5}}$ and to the temperature $T$) as inputs for various approaches and methods to improve the prediction of these regions. These methods include a Bayesian approach and different regression models. The data of the variables/proxies come from ERA5, the observation comes from MOZAIC/IAGOS, which we assume as

the truth. With the data of MOZAIC/IAGOS we can make the distinction between ISS and $\overline{\mathrm{ISS}}$. The evaluation of the different methods is carried out with a score value (ETS) that checks how well the prediction matches the observation.

To find out which dynamical variables are best suited for the regressions and which do not provide redundant information, we use various methods of information theory and test them. These include the log-likelihood ratios, known from the Bayes' theorem, a modified form of the Kullback-Leibler divergence, which we call the expectation of the absolute logit, and the

mutual information.

Log-likelihood ratios with values $> 2$ indicate that ISS is more likely than $\overline{\mathrm{ISS}}$. Only $RHi_{\mathrm{ERA5}}$ delivers values above 2 in a larger range, which indicates that ISS is more likely there than $\overline{\mathrm{ISS}}$. The vertical velocity $\omega$ and the relative vorticity $\zeta$ show also values above 2, but in a very small range. All other log-likelihood ratios are always below 2, which means that their effect on updating the prior odds ratio is quite small.

Particularly high values of the expectation for absolute logits are found for $RHi_{\mathrm{ERA5}}$, the lapse rate $\gamma$ and the relative vorticity $\zeta$, which means that these proxies have the greatest learning effect when assessing whether ISS or $\overline{\mathrm{ISS}}$.

Furthermore, to estimate the suitability of a proxy for a regression, we use the mutual information, which is a measure of how much information one variable $X$ can provide about another variable $Y$. To be a good predictor of ISS, it is important that the variable is both very well correlated with the relative humidity of the MOZAIC/IAGOS data, $RHi_{\mathrm{M/I}}$ (which we assume

as the truth), and at the same time as uncorrelated as possible with the other variables. $RHi_{\mathrm{ERA5}}$, $PV$, $\gamma$ and $\zeta$ have the highest mutual information to $RHi_{\mathrm{M/I}}$ but especially $PV$ and $\gamma$ are themselves quite strongly correlated, so it is sufficient for a regression to use $PV$ and omit $\gamma$ because of the higher mutual information of $PV$ with $RHi_{\mathrm{M/I}}$.

We use the most promising variables in several regression models to predict ISS. Through the regressions we find out that no matter which and how many dynamical proxies are added as an input, they do only provide little new information regarding

ISS. With only the raw $RHi_{\mathrm{ERA5}}$ and $T$ data of ERA5, Gierens et al. (2020) could achieve an ETS value of $0.08$ for the prediction of ice supersaturation. For the present data set it is $0.198$. The best regression that we can find, achieves an ETS of $0.378$. We consider this as not satisfying for flight routing.

It turns out that the dynamical proxies hardly provide information to the question of ISS or $\overline{\mathrm{ISS}}$, although the mutual information between in particular $RHi_{\mathrm{ERA5}}$ and $RHi_{\mathrm{M/I}}$ and between $PV$ and $RHi_{\mathrm{M/I}}$ is quite large. Furthermore, we see

that of all variables only $RHi_{\mathrm{ERA5}}$ has a range of values with the logit function exceeding the critical level of 2 and that a regression without $RHi_{\mathrm{ERA5}}$ is not successful. Thus, it turns out that the relative humidity $RHi_{\mathrm{ERA5}}$ at flight level is essential in our regressions in spite of its weaknesses. We suspect, that the main problem with predicting ISS is the strong overlap of the conditional probability density functions $P(RHi_{\mathrm{ERA5}}|ISS)$ and $P(RHi_{\mathrm{ERA5}}|\overline{ISS})$, especially in the critical region around $RHi_{\mathrm{ERA5}} = 70\%$ to $100\%$. Sensitivity tests show that the ETS increases strongly with a decrease in the overlap, which means

that if $P(RHi_{\mathrm{ERA5}}|ISS)$ and $P(RHi_{\mathrm{ERA5}}|\overline{ISS})$ were just slightly more separated, the results would drastically improve.



While corrections of $RHi_{\mathrm{ERA5}}$ lead to better predictions of ice supersaturation (increase in the ETS values) for comparing $RHi_{\mathrm{M/I}}$ with the modified relative humidities, they only slightly improve the forecast of potential persistent contrails (based on the ETS of the ISS-prediction) using regression methods, the proxies that turned out to be the most suitable ones and the modified relative humidities. This is due to the fact that also the overlap of the conditional PDFs $P(RHi|ISS)$ and
$P(RHi|\overline{ISS})$ of the modified relative humidities is hardly reduced, and probably also because the corrections are absorbed by the regressions and thus they do not become effective.

In the present paper we have used the meteorological data only at the point and time where the prediction of ice supersaturation is required. One can increase the effort and use additionally forecast data from earlier points in time and locations upstream of the location of interest (e.g. Duda and Minnis, 2009a, b). Wang et al. (2022) report that the humidity forecast of the
ECMWF model can be improved by application of a random forest fed with data from previous atmospheric states (a couple of hours) and covering about 100 hPa in vertical distance, in order to account for the past vertical motion that led to the current state. While this is certainly a possibility to improve the predicted humidity field *per se*, it is not clear whether the methods are fast and accurate enough for flight planning. Duda and Minnis (2009b) conclude that "reductions in the uncertainties of meteorological variables to a point where acceptable contrail forecasts are produced would be a good goal for NWA (numerical
weather analysis) modelers".

We conclude that the representation of $RHi$ in models of numerical weather prediction need to be improved. There are several ways to do this, but it will take some time to realise it. Cloud physics in numerical weather models is greatly simplified. This was justified for a long time because of the constraints of computing time and because the processes at flight level were not in the focus of weather prediction. However, as aviation needs to reduce its climate impact, avoidance of contrails gets
interesting for airlines and thus the prediction of ice supersaturation needs improvement. Furthermore, computer power is rising and additional resources can be used to improve the description of physical processes. A recent example is the concept of a one-moment scheme by Sperber and Gierens (2023). Furthermore, we think that more aircraft need to be equipped with hygrometers to measure humidity at flight altitudes for data assimilation. This would enable numerical weather models to improve the prediction of relative humidity for flight routing.

*Code availability.* R codes can be shared on request.

*Data availability.* ERA5 data can be obtained from the Copernicus Climate Data Service at https://cds.climate.copernicus.eu. MOZAIC/I-AGOS data are available from https://iagos.aeris-data.fr.



*Author contributions.* This paper is part of Sina Hofer's PhD thesis. SH wrote the codes, ran the calculations, analysed the results and produced the figures. KG supervises her research. Both authors discussed the methods and results and wrote the paper. SR curated the MOZAIC/IAGOS data.


*Competing interests.* The authors declare no competing interests.

*Acknowledgements.* The research described in this paper contributes to the Horizon 2020 project ACACIA (Grant No. 875036). The authors would like to thank Prof. Robert Sausen for the helpful discussion and Johanna Mayer for her thorough reading and commenting a draft manuscript.



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
