# Peer review of "How well can persistent contrails be predicted? — An update"

_EGUsphere, 2024_

## Author Comment (AC1)

**Subject: How well can persistent contrails be predicted? — An update**

*We thank the reviewers for the comments. For convenience, we repeat the comments and then give our replies, which are printed in italics.*

**Reply to referee #1**

This paper explores various approaches to improve the prediction of ice supersaturated regions that is provided by numerical weather prediction models. This piece of work is crucial in informing ongoing efforts and trials in attempt to mitigate aviation's contrail climate forcing. The paper is well-structured, employs suitable methodology, and presents reasonable results. It aligns with the scope of Atmospheric Chemistry and Physics. I recommend it for publication pending consideration of the following methodological questions and suggestions for improvements:

1. [Section 2.1] The description of the MOZAIC/IAGOS dataset might create confusion for the reader. For example, MOZAIC was transferred to the European infrastructure IAGOS in 2011, and the authors stated that only data between 2000 and 2009 was used. If this is the case, why not just name the in-situ measurements as "MOZAIC", instead of "MOZAIC/IAGOS"? In addition, what is the rationale for not including more in-situ measurements from the recent IAGOS dataset (i.e., between 2009 – present day)?

   *REPLY: Some people are more familiar with the term "IAGOS" and some with the term "MOZAIC". We use the term "MOZAIC/IAGOS" so that everyone knows which data is meant. We only use the data from 2000-2009 because we already had these available and also other studies have already been carried out with this data, such as:*

   *DOI: 10.3390/aerospace8110332,*

   *DOI: 10.5194/acp-22-7699-2022,*

   *DOI: 10.1127/metz/2024/1178.*

   For further studies we plan to extend the dataset to more recent data.

2. [Section 2.1] What is the temporal resolution (between flight waypoints) that is provided by the MOZAIC/IAGOS in-situ measurements? Have the authors considered and attempted to minimize the potential for autocorrelation between waypoints?

   *REPLY: You're right, we didn't describe this in detail in our paper. We will add a few explanatory sentences (in blue). MOZAIC/IAGOS in-situ measurements are available every four seconds. This corresponds to a flight distance of around one kilometer. We randomly selected only one percent of the data points (around every 100th point) to avoid autocorrelation. This means the temporal distance is around 400 seconds on average. Note that the average flight path length through ISSRs is around 150 km on average.*

3. [Section 2.2] It is unclear what is the specific product of the ERA5 reanalysis data that was used in this manuscript. Is it the ERA5 high resolution realization (HRES) reanalysis? In addition, what is the spatiotemporal resolution of the ERA5 data that was downloaded and used in this analysis? How does the spatiotemporal resolution of the meteorological data influence the results presented in this paper?

*REPLY: The hourly ERA5 high resolution realisation (HRES) reanalysis is used in this study. The spatial resolution of these data is 1° x 1°. We will add this to our paper. It has already been established in an earlier work that the differences between the nearest neighbor approach and a 4D interpolation are very small. In addition, a higher resolution probably does not lead to better results due to the lack of data and data assimilation.*

4. [Section 2.2] Given that the spatiotemporal resolution of the meteorological data is an important factor that can influence the quality of ISSR and RHi estimates, have the authors considered re-running the analysis using "model level" data in addition to "pressure levels"?

*REPLY: In this paper and in the former papers we preferred pressure levels because they are easier to handle than model levels because the pressure on model levels is different in every grid box and every time step. We don't believe that there is more reality with respect to the relative humidity field on model levels than on pressure levels because there is not enough measurement data for assimilation in the required resolution. We assume therefore that the decision whether model level or pressure level has no significant effect on our results. In addition, the preparation of the 10-year data in model level would be very time-consuming.*

5. [Section 2.2] It is worth including some details on the specific parameterization used to estimate the saturation pressure over ice ($p_{ice}$), which is required to estimate the RHi. Does the use of different parameterizations, i.e., Sonntag (1994) or Murphy & Koop (2005) lead to differences in the presented results?

*REPLY: Yes, we didn't mention the parameterisation we used in our study. For the calculation of the relative humidity with respect to water, the saturation vapour pressure over liquid water and over ice are calculated using the equation by Murphy and Koop (2005). Other parameterisations would probably lead to very slightly different results. The deviation of saturation pressure of water vapour over liquid water by Murphy and Koop from Sonntag's formula, for example, is less than 5% from 0 to -60° Celsius (see https://sciencepolicy.colorado.edu/~voemel/vp.html, Fig.2). We assume, that the effect is too small and that this reflects the noise.*

6. [Section 2.2] A recent open-source repository found that the interpolation method across the vertical level (i.e., linear interpolation, log-log interpolation, or cubic spline interpolation) could lead to differences in the RHi estimates from the ERA5 humidity fields (https://py.contrails.org/notebooks/specific-humidity-interpolation.html). For example, given the non-linear lapse rate of the specific humidity, a linear interpolation across the vertical level may lead to overestimation of the specific humidity.

Therefore, the authors should consider exploring the impact of the interpolation methodology on their presented results.

*REPLY: This question is very relevant for the interpolation of q, but not for RH. We do not calculate RH using q, but we interpolate RH directly by quadrilinear 4D interpolation.*

7. [General comment] The authors correctly highlighted that contrail mitigation is most effective when long-lived persistent (warming) contrails are avoided. Since contrails forming near the RHi threshold (close to 100%) tend to be shorter lived relative to those formed at higher RHi's, one potential strategy is to focus on regions where ice supersaturations are notably higher than threshold conditions. For instance, could there be an improvement in the ETS scores if the authors consider raising the threshold for the "predicted contrail formation" to be, say, $RHi_{ERA5} > 110\%$, instead of 100%?

*REPLY: This is a very good question. For a RHi threshold of 100% the a priori probability for ISS is 12.5%. For a higher RHi threshold of 110% the a priori probability for ISS is reduced to 6.7%. So, the logarithm of the prior odds ratio is then: $ln(0.067/(1 − 0.067)) \approx -2,6$. This means the log-likelihood ratios (logits) have not only to be greater than 2, but even greater than 2.6. This no longer works with dynamic proxies. You can see that quite clearly in the scatterplot below. $RHi_{ERA5}$ is plotted on the x-axis and $RHi_{M/I}$ is plotted on the y-axis. The blue dashed lines symbolise the humidity thresholds for 100 and 110%. For a better overview, not all data is shown, but only 500 randomly selected points. You can clearly see here that at 110% there is almost no data left in ERA5, while there are still a lot of points above 110% in MOZAIC/IAGOS. Therefore, in spite of this very good idea it does not work with our data.*

[Figure]

8. [Structure] Section 3 of the manuscript includes both the methodology and results, thereby making the section unnecessarily long. The authors should consider separating the methodology and results into two different sections to improve the readability of the manuscript.

   *REPLY: Thank you very much for this suggestion. But we would like to keep this structure, otherwise each method would be explained in the theory first and all results would be listed at the end. We think that our version prevents jumping back and forth in the text and the reading flow would be improved if you could first read the theory and then the results for each individual method. In addition, the connections and conclusions that we have drawn from it become clearer.*

**Reply to referee #2**

This paper examines the key challenge of accurately predicting ice supersaturated regions (ISSRs) at cruising altitudes using reanalysis data and in situ measurements as ground truth. This paper is a continuation and culmination of some of the authors' efforts and previous research to use dynamical proxies to improve the ability to predict relative humidity with respect to ice. It aims to provide an end-to-end statistical framework for better estimation of ISSR formation, focusing on the different statistical design options and their respective performance. The conclusion, although disappointing, is a significant result for the contrail research community.

The paper is well organised, uses appropriate statistical methods and provides plausible and important results. We recommend its publication as it is, with only a minor technical correction:

[Section 3.1] "This means that the likelihood ratio must be greater than 3.85". I assume here that 3.85=0.5/0.13. But if I'm not mistaken, to make ISS more likely than no ISS, we need the posterior odds ratio to be greater than 1.0, which translates into a likelihood ratio threshold of 1.0/0.13=7.69. Since ln(7.69)=2.0, this would be consistent with the threshold on the logit

*REPLY: Yes, you are absolutely right. We made a mistake at this point. Since the a-posteriori ratio must be greater than 1 to make ISS more probable than no ISS and the a priori probability for ISS in our data is 0.125 (and not 0.115 as we wrote in the paper) the likelihood ratio has to be 1.0/0.14 = 7.14.*

Here are also some points to consider for further scientific discussion or future research, particularly with regard to the statistical methodology (Part 3):

1. [Section 2.1] As mentioned by the other reviewer, it would be interesting to understand how the spatio-temporal resolution of the meteorological data affects the statistical results developed in the Part 3.

   *REPLY: See reply 2 to question 2 from referee 1.*

2. [Section 3.1] It could be interesting to use the same validation framework (split train/test datasets) to estimate the ETS using the individual Bayesian estimators directly as binary classifiers. Section 3.1 could be developed in a very similar way to 3.2, as methods based on Kullback-Leibler distance can be used as model/variable selectors (although less frequently than mutual information). Distance between variables is not needed if you are using forward additive selection or backward elimination.

   *REPLY: That's a good idea we could think about in the future. But for this paper it would widen the scope and perhaps therefore it would distract the reader from the main line of reasoning.*

3. [Section 3.1] It might be interesting to use slightly more complex Bayesian estimators such as naive Bayes, which tend to work surprisingly well even in a situation where the variables are correlated, provided it is improved by optimising the threshold applied to the posterior output. Accounting for variable correlation with a simplified DAG and simple conditional probability parameterisations is also an option.

   *REPLY: According to our results from the mutual information, we know that the dynamical proxies are not really independent. Therefore, we are uncertain what the naive Bayes would tell us. It would be certainly interesting to test it but we keep it for future studies.*

4. [Section 3.2.2] The models chosen in this section could be automatically determined by forward additive selection or backward elimination using the mutual information as variable selector.

   *REPLY: We agree.*

5. [Section 3.2.4] Ridge or lasso regularisation could be used to better control the risk of overfitting due to correlated variables.

   *REPLY: We agree. We think it does not really matter for our problem since we could show that the results do not get very much better whether we include the proxies or not.*

6. [Section 3.2.4] Although it is highly unlikely that it would have changed the results at all, modern tree-based machine learning techniques (random forests/LightGBM) could be used as they are designed to deal with non-linearities, overfitting/correlated variable problems and implicit variable selection at the same time.

   *REPLY: We agree.*

7. [Section 3.2.4] Optimising the output probability threshold might improve the results slightly (but certainly not significantly).

   *REPLY: We tested different thresholds for our methods and we came to the result that for 0.34 the ETS is maximal (see the three last lines before chapter 3.2.4).*

8.  [Section 4.2.3] As mentioned by the authors, it is not surprising that adjustment techniques like [Teoh et al., 2022] are not really needed here, as the non-linear nature of the classifier has learned to infer the implicit bias non-parametrically.

    *REPLY: We agree and are happy that you confirm this.*

9.  [General] As mentioned by the other reviewer, it might be interesting to reformulate the problem as a multi-class problem, either using situations of even higher supersaturation or using the ontology developed in [Wilhelm et al., 2020] (no persistent contrails, persistent contrails, most warming contrails). A direct regression framework between reanalysis and in-situ relative humidity is also an option.

    *REPLY: Unfortunately, it does not really work. See reply to the final question of referee 1.*